# Farm Size, Risk Aversion and Overuse of Fertilizer: The Heterogeneity of Large-Scale and Small-Scale Wheat Farmers in Northern China

**Haixia Wu [1], Hantao Hao [1], Hongzhen Lei [1], Yan Ge [2,\*], Hengtong Shi [1] and Yan Song [1]**

[1] International Business School, Shaanxi Normal University, Xi'an 710119, China; haixiawu@snnu.edu.cn (H.W.); 15081607693@163.com (H.H.); leihongzhen@snnu.edu.cn (H.L.); shihengtong@snnu.edu.cn (H.S.); chaliesong@snnu.edu.cn (Y.S.)

[2] School of Public Finance and Tax, Central University of Finance and Economics, Beijing 100081, China

\* Correspondence: yge001@ucr.edu

**Abstract:** The excessive use of fertilizer has resulted in serious environmental degradation and a high health cost in China. Understanding the reasons for the overuse of fertilizer is critical to the sustainable development of Chinese agriculture, and large-scale operation is considered as one of the measures to deal with the excessive fertilizer use. Under the premise of fully considering the resource endowment and heterogeneity of large-scale farmers and small-scale farmers in production and management, different production decision-making frameworks were constructed. Based on the 300 large-scale farmers and 480 small-scale farmers in eight provinces of northern China wheat region, we analyzed the optimal fertilizer use amount and its deviation as well as the influencing factors of small-scale and large-scale farmers, then further clarified whether the development of scale management could solve the problem of excessive fertilizer use. The empirical results show that: (1) both small-scale farmers and large-scale farmers deviated from the optimal fertilizer application amount, where the deviation degree of optimal fertilizer application of small-scale farmers is significantly higher than that of large-scale farmers, with a deviation degree of 35.43% and 23.69% for small and large scale farmers, respectively; (2) not all wheat growers in North China had the problem of excessive use of chemical fertilizer, as the optimal level of chemical fertilizer application in Heilongjiang and Inner Mongolia are 346.5 kgha$^{-1}$ and 335.25 kgha$^{-1}$, while the actual fertilizer use amount was 337.2 kgha$^{-1}$ and 324.6 kgha$^{-1}$, respectively; and (3) the higher the risk aversion level, farmers tended to apply more fertilizer to ensure grain output. Therefore, increasing farm size should be integrated into actions such as improving technological innovation and providing better information transfer to achieve the goal of zero-increase in Chinese fertilizer use.

**Keywords:** scale management; risk aversion; optimal fertilizer use; environmental protection





## 1. Introduction

Feeding a growing and increasing wealthy population is a major challenge. Since 2003, China's grain production has made great achievements in the twelfth consecutive increase under the stimulation of economic development, population explosion, technological progress, and policy support [1–3]. On the other hand, the traditional dual structure of urban and rural areas in China has brought about the reduction of cultivated land area and the transfer of the rural labor force, resulting in the shortage of cultivated land resources and rural labor forces [3,4]. In this context, in order to promote food supply and ensure food security, the "high-input-high-output" agricultural production model has been adopted by many regions and popularized on a large scale, as the main feature of this model is to increase the input of chemical fertilizers to achieve high output [5]. For a long time, the overuse of fertilizer has been widely suspected by small-scale farmers in grain production, and it has been considered as one of the main causes of aggravation of agricultural non-point source pollution [6–8].

Data released by the Ministry of Agriculture in 2017 showed that the average amount of fertilizer applied to crops in China was 328.5 kgha$^{-1}$, far higher than the world average of 120 kgha$^{-1}$, 2.6 times and 2.5 times that of the United States and the European Union, respectively. Excessive application of fertilizer has directly led to the significant reduction of fertilizer utilization rate of cultivated land in China. In 2017, the average soil organic matter of cultivated land in China was less than 1%, and the utilization rates of nitrogen, phosphorus, and potassium fertilizers for the three major grain crops were only 33%, 24% and 42%, respectively. This not only caused huge economic losses, but also brought serious pollution to the environment, water, soil, atmosphere, biology, and human health [4,9,10].

Interestingly, in recent years, replacing small farms with bigger ones seems to have become a popular solution among researchers to address the overuse of agricultural chemical fertilizer worldwide [11–14]. The promotion of large farms is in line with the mainstream claims on global agriculture [15–18]. Some authors blame the small size of Chinese farms for the overuse of chemicals and believe that scale producers only pursue profit maximization, there is no reason to overuse the chemical fertilizer [12,19]. Based on this acknowledgement, numerous researchers recommend various forms of land consolidation to reduce the fertilizer application and resulting ecological damages [12,20,21]. Meanwhile, researchers have come to an agreement that due to mastering more capital and scientific knowledge, larger farms are more able to exploit precise fertilization technologies and management, which results in the significant influence of land scale on fertilizer usage amount [22–24]. For instance, Ju et al. [21] found that the scale of farm land management had a significant effect on the decrease in fertilizer application rate in China. For every 1% increase of the average cultivated land area, the fertilizer application rate per hectare will decrease by 0.3%. Other research on farm size and productivity seem to suggest that larger farms tend to enjoy economies of scale and have higher efficiency and lower fertilizer intensity or better sustainability, and this opinion is often echoed by international mainstream media [25,26].

However, whether for small farmers the investment of fertilizer is really "irrational"or "excessive" or whether scaling cultivation can effectively avoid the excessive use of chemical fertilizers is still being argued by many scholars. We have compared and illustrated the differences of decision-making motivation, behavior, and equilibrium results of fertilizer use between large-scale households and small-scale households. Based on the survey data of wheat scale producers and ordinary farmers in eight provinces of the northern wheat region including Heilongjiang, Inner Mongolia, Shanxi, Shaanxi, Gansu, Ningxia, Qinghai, and Xinjiang, the optimal fertilizer application of wheat production was calculated, and the deviation of fertilizer application amount between a large-scale household and ordinary household was compared. On this basis, the main contributing factors leading to the deviation of fertilizer application between large-scale households and ordinary households were analyzed.

The contributions of this paper have two aspects. First, it fully considers the differences between small-scale farmers and large-scale farmers through their own resource endowment and production and management purposes, and the theoretical analysis frameworks based on family utility maximization and profit maximization are constructed to investigate the optimal fertilizer application and its deviation for small-scale and large-scale farmers. Second, it uses empirical analysis to investigate the main factors causing fertilizer application deviation and to provide a new understanding of the differences in the fertilizer application behavior of farmers in China.

## 2. Analysis Framework of Fertilizer Use Behavior between Large-Scale Farmers and Small-Scale Farmers

As an independent economic entity, the behavioral goal of farmers is to maximize utility or income [27–29]. In theory, only when the utility of a commodity or service can be represented by an undifferentiated value or price, can income maximization be equivalent to utility maximization. If the utility cannot be equal to the price, for instance, when there is a risk, the utility of different goods or services is different, and the decision maker

will not only pursue income maximization. However, at this time, their behavior is still rational, and it can even be said to be a more rational choice. Large-scale households are specialized organizations for agricultural production, and their business activities are in full compliance with the theory of the firm, and their production activities seek to maximize their profits. This means that the production of farmers is always at the point where the marginal cost equals the marginal revenue.

Taking into account the factor market, under equilibrium conditions, the marginal value of each product factor input by farmers must be equal to the price of the factor, and the marginal value of the product factor is equal to the multiplication of marginal output and the price, so as the input of factors increases, the marginal value of the product diminishes. Figure 1 indicates the relationship between fertilizer input and marginal product value. When other factor inputs remain unchanged, increasing chemical fertilizer input will result in a decrease in the output of the marginal product value of crops. Suppose that the marginal product output value curve of large-scale households is $VMP_L$ and the fertilizer price at the market is $PF$, the optimal fertilizer input for large-scale households should be $L^*$. However, in reality, the actual application amount of chemical fertilizers for large-scale households may be higher than this optimal value because chemical fertilizer application has a practical problem that is different from the general agricultural input of elements. Fertilization application is closely related to the original soil fertility and the content of chemical fertilizers. Farmers in China often have insufficient knowledge of soil fertility and fertilizer nitrogen, phosphorus, and potassium content information, and large-scale households are no exception, especially large-scale households whose main land is rented from the government, and even have less information regarding the land than ordinary households. Therefore, due to risk aversion considerations, both large-scale and small-scale households may have a tendency to apply more fertilizers [30–32].

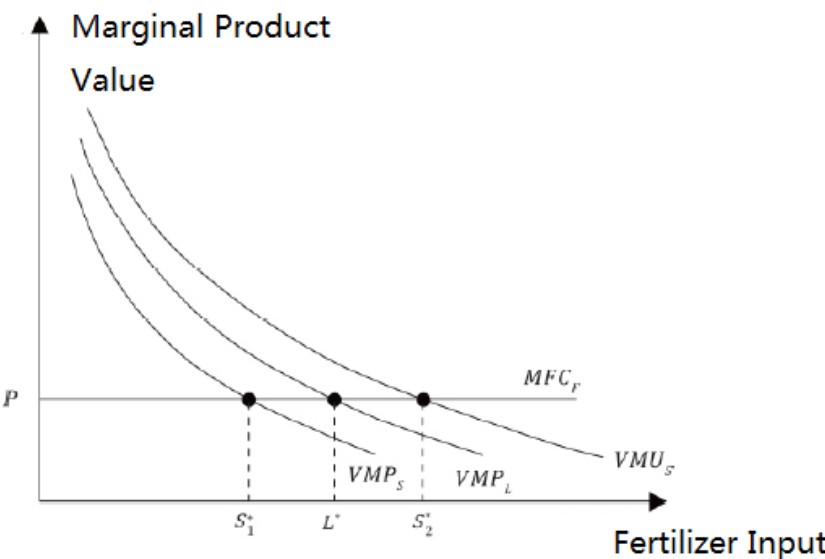

**Figure 1.** The relationship between fertilizer input and marginal product value.

However, as most small-scale households follow the semi-self-sufficiency model, the decision first needs to satisfy the self-sufficient consumption needs, and then seek to maximize family utility. Although small-scale households mainly rely on non-agricultural income for their family income, however, there is a high extent to which they rely on self-produced food and they pay more attention to ensuring family food security through the quantity and quality of self-produced food. Therefore, the utility brought by each unit of product cannot be simply measured by its market price, and the marginal product value curve corresponding to small-scale households should include two parts: the marginal product economic value and the household food safety utility value.

Similarly, taking the input of chemical fertilizers as an example, when the profit is maximized, the output cannot meet the family needs. In order to ensure the safety of family food, the farmers have to increase the output by continuing to increase the fertilizer to meet their own needs. As shown in Figure 1, assume that when only small-scale households pursue profit maximization, the marginal product output curve is $VMP_s$, and the optimal amount of fertilizer application is $S_1^*$. When household food safety utility is taken into account, the marginal product value curve shifts to the right from $VMP_s$ to $VMU_s$. Under the same fertilizer price, the optimal amount of fertilizer application is $S_2^*$. The change from $S_1^*$ to $S_2^*$ can be understood as that when the yield at profit maximization is smaller than the household demand, in order to ensure food security, farmers will sacrifice efficiency and profit to increase the yield (i.e., in the utility function, when the application amount of fertilizer exceeds $S_1^*$, its marginal product utility is still higher than the unit factor price, although its marginal product output value is less than the unit factor price). Therefore, the factor input will continue to increase until $S_2^*$.

The marginal product utility of the last unit of fertilizer input is equal to the fertilizer factor price. Therefore, compared with the traditional assumptions that small-scale households pursue the way of fertilizer application for profit maximization, in reality, ordinary households will systematically have an excessive application of chemical fertilizer. In addition, like large-scale households, risk aversion further intensifies the degree of excessive fertilizer use in ordinary households. As a result, it is possible to see in practice that the average small-scale household uses more fertilizer per hectare than the average scaled household. However, this is not an irrational behavior, because under the constraints of specific resources and environment, small-scale households pursue utility maximization through preference selection in order to ensure the food safety of the family.

According to the above analysis, this paper proposes the following research hypotheses regarding the effect of scales and risks on fertilizer application and its deviation:

**Hypothesis 1:** *Since small-scale households pursue maximum utility in grain production, compared with large-scale grain production households, small-scale households have a greater deviation from the optimal fertilizer application.*

**Hypothesis 2:** *Due to insufficient information on soil fertility and effective components of chemical fertilizers, households also have a risk aversion tendency. Fertilizer application will deviate from the economic optimal level. The higher the risk aversion degree, the greater the deviation of chemical fertilizer application by the household.*

Previous studies have shown that the risk aversion of households has a direct impact on their fertilizer application behavior, and the degree of risk aversion affects their agricultural production and input behavior [33]. Small-scale households have a higher degree of risk aversion, which helps to explain some of the households' production and operation behaviors deviating from the profit maximization goal. For example, Qiu et al. (2014) [31] pointed out that the households' cultivated land area was significantly negatively correlated with fertilizer input, and they avoided agricultural production risks by increasing the application amount of fertilizer. Moreover, the bigger the scale of cultivated land, however, the greater percentage of income derived from agriculture by households, hence they are more willing to learn the technical operation methodologies of chemical fertilizers and pesticides through various channels, and actively learn from the successful experience and failure lessons [34] so as to apply the appropriate amount of fertilizer to rationalize the long-term development of agricultural production. On this basis, this paper proposes the third research hypothesis:

**Hypothesis 3:** *The scale of cultivated land can mitigate the behavior of fertilizer application due to risk aversion by households.*

## 3. Materials and Methods

In order to verify the hypothesis above, this paper first conducted descriptive statistics on the data by analyzing and comparing the difference in fertilizer use between large-scale and ordinary households, and then estimated the output elasticity of the two main types of fertilizers through the estimation of the production function, then based on which the optimal application rate was calculated. By comparing the actual fertilizer application rate, we observed whether there was a deviation in the chemical fertilizer application. Finally, by calculating the difference between the optimal application rate and the actual application rate, the deviation degree of the farmer's fertilizer application was obtained, based on which we established a measurement model to analyze the influencing factors.

### 3.1. Model Settings

In economics, the profit is maximized when the marginal product value of the input factor is equal to its price. Therefore, by comparing whether the ratio of the marginal output value of chemical fertilizers to the price of chemical fertilizers is less than one 1, we can judge whether there is a deviation in the application of chemical fertilizers.

First of all, on the basis of referring to previous studies, this paper chose the logarithmic Cobb–Douglas production function model to measure the output elasticity of fertilizer:

$$\ln Y = \beta_0 + \beta_1 \times \ln F + \beta_2 \times \ln L + \beta_3 \times \ln M + \beta_4 \times \ln S + \beta_i \times Farmer + \beta_j \times Land + \varepsilon_1 \tag{1}$$

where the dependent variable $Y$ represents the wheat yield per hectare, and the independent variables include the input of production factors such as fertilizer input ($F$, indicating the amount of fertilizer per hectare), labor input ($L$, indicating days needed per hectare), machinery input ($M$, indicating the cost of machinery input per hectare), seed input ($S$, indicating the cost of seed input per hectare), and personal characteristics of the farmer (*Farmer* including the gender and education level of the head of the household, etc.), land plot characteristics (*Land* such as whether technology demonstration field, whether other crops have been planted, and the soil quality of the plot, etc.). $\varepsilon_1$ represents a random disturbance term. The output elasticity of fertilizer is $\beta_1$.

Furthermore, based on the principle of profit maximization, we can calculate the optimal amount of fertilizer for farmers. The condition for farmers to obtain the maximum benefit is that the marginal benefit is equal to the marginal cost, that is, the marginal effect of chemical fertilizers on wheat production should be equal to the ratio of chemical fertilizer prices to wheat prices. At the same time, under the condition of the marginal effect of chemical fertilizers on wheat yield calculated based on Formula (1), the optimal application amount of chemical fertilizer per hectare can be obtained as follows:

$$F_{optimal} = \frac{\beta_1 \times Y}{P_F / P_Y} \tag{2}$$

Finally, after calculating the optimal application rate of chemical fertilizer per hectare according to Formula (2), we can obtain the deviation degree of chemical fertilizer application per hectare by subtracting the optimal application rate from the actual application rate of chemical fertilizer per hectare.

$$F_{deviate} = F_{actual} - F_{optimal} \tag{3}$$

Since one of the objectives of the analysis in this paper was to compare the difference between large-scale households and ordinary households in deviating from the economic optimum in fertilizer application, the following part of the paper will construct the production functions of large-scale households and small-scale households separately to obtain the optimal amount of fertilizers application and the actual deviation.

In this paper, multiple regression method was used to analyze the influencing factors of the fertilizer application deviation degree of scale households and ordinary farmers. After controlling a series of influencing factors at the level of farmers and plots, we focused

on the differences between scale households and ordinary farmers in the deviation of fertilizer application from economic optimization. The model is specified as follows:

$$F_{deviate} = \alpha_0 + \alpha_1 \times Ifscale + \alpha_2 \times Risk + \alpha_3 \times Farmer + \alpha_4 \times Land + \alpha_5 \times Other + \varepsilon_2 \quad (4)$$

where the dependent variable $F_{deviate}$ represents the degree of deviation of the fertilizer application by farmers on a certain plot. $Ifscale$ and $Risk$ are the key explanatory variables in this article, representing the risk aversion degree of large-scale households and farmers, respectively. $Other$ indicates the other factors that may affect farmers' chemical fertilizer application behaviors controlled by previous studies such as social capital, whether the family members have served as a village cadre, and the number of trainings attended. $\varepsilon_2$ is the residual term, which represents the unobservable factors not covered by the existing variables.

### 3.2. Data Description

The data used were from the survey data of 900 households in 64 villages and 32 towns in 16 counties of Heilongjiang, Inner Mongolia, Shanxi, Shaanxi, Gansu, Ningxia, Qinghai, and Xinjiang from August to October, 2020. After removing 120 invalid questionnaires, 780 valid questionnaires were collected including 300 large-scale farmers and 480 small-scale farmers. What is clear is that smallholder farms were typically less than 2 ha in this paper, although the definition of smallholder used in national censuses varies considerably [35]. At the same time, it should be noted that Heilongjiang, Inner Mongolia, and Xinjiang have the largest per capita cultivated land area in China, and most of the wheat growers in this area are large-scale farmers, while Shanxi, Shanxi, and Gansu are typical small-scale farmers. Considering the representativeness of the samples, in terms of sample allocation, first, according to the wheat sown area of each province in 2019, the distribution of scale growers in each province was determined, and multi-stage random sampling was carried out in each province in combination with the local economic development level, regional distribution of wheat planting, and distribution of scale operators. Then, 20 ordinary wheat growers were randomly selected around the large-scale farmers for questionnaire survey. The sample distribution of small-scale and large-scale wheat farmers in each province is shown in Table 1.

**Table 1.** Provincial distribution of small-scale and large-scale farmers (unit: households).

| Province | Heilongjiang | Inner Mongolia | Shanxi | Shaanxi | Gansu | Ningxia | Qinghai | Xinjiang |
|---|---|---|---|---|---|---|---|---|
| Small-scale | 66 | 64 | 51 | 72 | 53 | 34 | 32 | 108 |
| Large-scale | 41 | 40 | 32 | 45 | 33 | 21 | 20 | 68 |

Table 2 presents the variables used in this paper, the description of variables, and their statistical characteristics. It can be seen that the average yield of wheat in the north of China is 6349.05 kgha$^{-1}$, and the input of chemical fertilizer is 681.30 kgha$^{-1}$. The average degree of risk aversion is 1.42, which indicates that the degree of risk aversion is higher among the sampling farmers. The annual per capita net income of the family is 11,400 Yuan, and the non-farm income accounts for 32.45% of the total income of the family. According to the characteristics of wheat plots, the average plots are 2.45, and most plots are crop rotation, and the site type is mainly sloping land.

**Table 2.** Variable description and statistical description.

| Variables | Abbreviations | Description | Mean | S.D. |
|---|---|---|---|---|
| Wheat yield per hectare | Yield | kgha$^{-1}$ | 6349.05 | 1535.40 |
| Large-scale farmer | Scale | 1 = Yes, 0 = No | 0.38 | 0.49 |
| Application amount of chemical fertilizer | Fertilizer | kgha$^{-1}$ | 681.30 | 181.35 |
| Seed input | Seed | Yuanha$^{-1}$ | 1332.70 | 409.95 |
| Machinery input | Machinery | Yuanha$^{-1}$ | 2651.40 | 714.6 |
| Labor input | Labor | Daysha$^{-1}$ | 29.08 | 4.16 |
| Irrigation input | Irrigation | Yuanha$^{-1}$ | 977.85 | 369.60 |
| Crop rotation | Rotation | 1 = Yes, 0 = No | 0.71 | 0.45 |
| Site type | Site | 1 = Flat land, 2 = Sloping land, 3 = Table land, 4 = Plateau | 2.11 | 0.76 |
| Number of lots | Lot | lots | 2.45 | 1.17 |
| technology demonstration farmer | Demonstration | 1 = Yes, 0 = No | 0.23 | 0.42 |
| Gender of Householder | Gender | 1 = Male, 0 = Female | 0.75 | 0.43 |
| Educational level of Householder | Education | Years | 7.02 | 2.55 |
| Wheat planting years | Years | Years | 27.33 | 14.21 |
| Annual per capita net income of family | Income | 10 Thousand Yuan | 1.14 | 1.77 |
| Proportion of non-agricultural income in total household income | Proportion | % | 32.45 | 28.34 |
| Number of labor | | | 3.59 | 1.28 |
| Do family members have party members, village or township cadres | Member | 1 = Yes, 0 = No | 0.26 | 0.44 |
| Training times of wheat planting technology in the past year | Training | Times | 2.40 | 0.69 |
| Farmers' risk aversion | Risk | You have been recommended a new agricultural production technology, but you don't know what the result will be. Would you like to adopt it? Even if someone uses it, I won't use = 1. If someone uses it, I'll use = 2. No matter whether someone uses it or not, I'll try to use = 3 | 1.42 | 0.88 |

Table 3 shows the input–output status of wheat production in northern China. Among the eight provinces in the northern wheat region, the difference of wheat yield per unit area was large. The highest yield provinces were Shanxi and Shaanxi, and the lowest yield provinces were Xinjiang, Inner Mongolia, and Heilongjiang. In terms of fertilizer input, Shanxi, Shaanxi, and Xinjiang were the provinces with a large amount of fertilizer input, while Inner Mongolia and Heilongjiang Provinces had lower input. In terms of seed, machinery input, and labor input, Shanxi and Shaanxi were also significantly higher than the other six provinces. In terms of irrigation input, due to the difference natural conditions, Heilongjiang, Inner Mongolia, Gansu, and Qinghai were basically dominated by natural precipitation, and the irrigation input was 0.

At the same time, Table 3 shows that the wheat yield of large-scale farmers was generally higher than that of small-scale farmers, and the amount of fertilizer, seed, and artificial application was low. Due to regional differences, the mechanical input of large-scale farmers was not lower than that of small-scale farmers in all provinces. It can be seen that a wheat planting mode with high input and high output generally exists in North China, whether it is large-scale farmers or small-scale farmers.

**Table 3.** Comparison of the input of production factors between small-scale and large-scale households in different provinces.

| Province | Farmers | Yield (kgha$^{-1}$) | Fertilizer (kgha$^{-1}$) | Seed (Yuanha$^{-1}$) | Machinery (Yuanha$^{-1}$) | Labor (Daysha$^{-1}$) | Irrigation (Yuanha$^{-1}$) |
|---|---|---|---|---|---|---|---|
| Heilongjiang | Large-scale | 5094.90 | 337.20 | 900.60 | 2702.10 | 18.60 | 0 |
| | Small-scale | 4292.10 | 420.30 | 1053.45 | 2859.60 | 34.05 | 0 |
| Inner Mongolia | Large-scale | 4930.35 | 324.60 | 949.35 | 2179.95 | 23.40 | 0 |
| | Small-scale | 4326.60 | 458.70 | 1132.20 | 1893.90 | 31.95 | 0 |
| Shanxi | Large-scale | 8700.15 | 649.35 | 1508.55 | 3247.2 | 27.45 | 754.80 |
| | Small-scale | 8317.05 | 909.75 | 1878.75 | 3013.65 | 38.40 | 1148.40 |
| Shaanxi | Large-scale | 8664.00 | 750.30 | 1530.75 | 2991.90 | 29.10 | 1152.15 |
| | Small-scale | 8222.40 | 984.60 | 1871.55 | 2809.50 | 39.75 | 1341.60 |
| Gansu | Large-scale | 7241.25 | 535.80 | 684.30 | 1501.35 | 17.25 | 0 |
| | Small-scale | 6453.30 | 607.35 | 877.65 | 2250.60 | 22.35 | 0 |
| Ningxia | Large-scale | 6680.55 | 495.15 | 1248.30 | 1805.55 | 18.30 | 907.05 |
| | Small-scale | 6495.90 | 574.20 | 1362.15 | 1774.20 | 24.60 | 949.95 |
| Qinghai | Large-scale | 6038.25 | 510.45 | 1186.95 | 2017.95 | 20.70 | 0 |
| | Small-scale | 5871.45 | 619.65 | 1385.85 | 1761.15 | 29.85 | 0 |
| Xinjiang | Large-scale | 5012.70 | 682.05 | 1152.15 | 2477.70 | 19.05 | 1204.05 |
| | Small-scale | 4335.75 | 759.30 | 1534.80 | 2251.50 | 24.15 | 1415.70 |

## 4. Results

### 4.1. Analysis of Factors Affecting Wheat Yield

We constructed ordinary least square (OLS hereinafter) models to analyze the influencing factors of wheat yield. As illustrated in Table 4, OLS regression results showed that the amount of chemical fertilizer had a significant positive impact on wheat yield. Specifically, the regression coefficient of large-scale farmers was 0.102, which indicates that an increase of 1% of chemical fertilizer application will increase the wheat yield of large-scale farmers by 0.102%. The regression coefficient of small-scale farmers was 0.128, which indicates that an increase in fertilizer application by 1% will increase the wheat yield of small-scale farmers by 0.128%. It can be seen that the output elasticity of the chemical fertilizer input of small-scale farmers in wheat production was greater. For both large-scale farmers and small-scale farmers, the site type had a negative impact on wheat yield, that is, in the transition from flat land to table land, and the wheat yield also showed a downward trend. The farmers who participated in the demonstration of wheat high-tech also had a higher wheat yield.

In terms of large-scale households, whether irrigation conditions were available or not had a significant impact on wheat yield, however, seed input, mechanical input, labor input, crop rotation, and the number of cultivated plots had no significant impact on wheat yield. For small-scale farmers, more mechanical input was helpful to improve the wheat yield, while the wheat yield of farmers with crop rotation was low. At the same time, the land fragmentation management was not conducive to the improvement of wheat yield. However, the effects of seed input, labor input, and irrigation conditions on wheat yield were not significant.

Moreover, from the perspective of control variables, the education level of the head of household and the years of wheat planting had significant positive effects on both large-scale and small-scale farmers, indicating that the higher the level of education and the more experienced farmers were, the more likely they were to achieve good planting output in wheat production. However, the net income per capita and the proportion of agricultural income in total household income had no significant effect on wheat yield.

**Table 4.** OLS regression results of influencing factors on wheat yield.

| Variables | Large-Scale Farmers | Small-Scale Farmers |
|---|---|---|
| Fertilizer | 0.102 *** | 0.128 *** |
| | (2.87) | (3.24) |
| Seed | 0.027 | 0.035 |
| | (0.49) | (1.24) |
| Machinery | 0.144 | 0.171 *** |
| | (3.96) | (4.22) |
| Labor | 0.015 | 0.042 |
| | (0.27) | (0.46) |
| Irrigation | 0.044 ** | 0.051 |
| | (2.16) | (1.28) |
| Rotation | −0.675 | −1.112 ** |
| | (−1.02) | (−2.09) |
| Site | −1.412 *** | −1.121 *** |
| | (−3.57) | (−4.88) |
| Lot | −0.615 | −0.421 * |
| | (−1.49) | (−1.72) |
| Demonstration | 0.513 *** | 0.394 *** |
| | (2.94) | (3.12) |
| Gender | 0.362 | 0.225 |
| | (0.49) | (0.56) |
| Education | 1.155 ** | 0.735 *** |
| | (2.13) | (2.78) |
| Years | 0.435 ** | 0.465 ** |
| | (1.95) | (1.97) |
| Income | 0.675 | 0.572 |
| | (1.57) | (1.29) |
| Proportion | 0.840 | 0.585 |
| | (1.24) | (1.43) |
| Training | 0.584 * | 0.855 ** |
| | (1.87) | (2.06) |
| Area dummy variable | Province | Province |
| Constant | 141.781 *** | 99.753 *** |
| | (3.88) | (4.57) |
| No. of Obs | 300 | 480 |
| $R^2$ | 0.494 | 0.533 |
| F Test | 21.63 | 24.39 |

Notes: * $p < 0.10$, ** $p < 0.05$, *** $p < 0.01$ level of significance.

### 4.2. Deviation Degree of Chemical Fertilizer Use

According to the OLS regression results of influencing factors of wheat yield in Table 4, the output elasticity of chemical fertilizer to large-scale growers and small-scale farmers were 0.102 and 0.128, respectively. According to Formula (2), the optimal fertilizer application amount for large-scale and small-scale farmers was 445.05 kgha$^{-1}$ and 499.2 kgha$^{-1}$, respectively. According to Formula (3), it can be calculated that the deviation degree of fertilizer application amount of large-scale farmers and small-scale farmers was 105.45 kgha$^{-1}$ and 176.85 kgha$^{-1}$, respectively, and the deviation rates were 23.69% and 35.43%, respectively. Meanwhile, the deviation degree *t* test in Table 5 is highly significant at the significance level of 1%, which indicates that both large-scale farmers and small-scale farmers had different degrees of deviation from the optimal fertilizer application amount. However, from the results, the deviation of fertilizer application rate from the optimal application rate was more significant.

**Table 5.** Deviation of fertilizer application between large-scale and small-scale farmers.

| Item | Large-Scale Farmers | Small-Scale Farmers |
|---|---|---|
| Actual amount of fertilizer application (kgha$^{-1}$) | 550.50 | 676.05 |
| Optimal amount of fertilizer application (kgha$^{-1}$) | 445.05 | 499.20 |
| Deviation degree of fertilizer application (kgha$^{-1}$) | 105.45 | 176.85 |
| Deviation rate (%) | 23.69 | 35.43 |
| *t* Test of deviation degree | 6.20 *** | 7.80 *** |

Notes: *** $p < 0.01$ level of significance.

### 4.3. Deviation Degree of Chemical Fertilizer Use in Different Provinces

Due to the large longitude span, significant differences in natural environment and complex climatic conditions, the application of chemical fertilizer in northern China demonstrated significant regional characteristics. Therefore, according to the regression results in Table 4 and Formulas (2) and (3), the deviations of fertilizer application of wheat farmers in different provinces were calculated, and the results are shown in Table 6. On the whole, wheat growers in different provinces of northern China had different degrees of fertilizer application deviation, and the deviation of small farmers was more obvious than that of large-scale farmers. Specifically, the deviation of fertilizer application was the most obvious among wheat growers in Xinjiang, with 341.25 kgha$^{-1}$ and 400.95 kgha$^{-1}$ for large-scale and small-scale farmers, respectively. The second was small-scale farmers in Shaanxi Province, and the deviation degree of fertilizer application was also very significant. The average fertilizer application amount per hectare deviated from the optimal application amount by 304.95 kgha$^{-1}$. However, it should be noted that the fertilizer application rates of large-scale farmers in Heilongjiang and Inner Mongolia were 337.20 kgha$^{-1}$ and 324.60 kgha$^{-1}$, respectively, which were the optimal levels of 346.50 kgha$^{-1}$ and 335.25 kgha$^{-1}$.

**Table 6.** Deviation of fertilizer use between large-scale and small-scale farmers in different provinces.

| Province | Farmers | Actual Amount of Fertilizer Application (kgha$^{-1}$) | Optimal Amount of Fertilizer Application (kgha$^{-1}$) | Deviation Degree of Fertilizer Application (kgha$^{-1}$) | Deviation Rate (%) | *t* Test of Deviation Degree |
|---|---|---|---|---|---|---|
| Heilongjiang | Large-scale | 337.20 | 346.50 | −9.30 | −2.68 | 1.021 |
| | Small-scale | 420.30 | 354.75 | 65.55 | 18.48 | 3.410 *** |
| Inner Mongolia | Large-scale | 324.60 | 335.25 | −10.65 | −3.18 | 1.482 |
| | Small-scale | 458.70 | 357.60 | 101.10 | 28.27 | 2.784 *** |
| Shanxi | Large-scale | 649.35 | 591.60 | 57.75 | 9.76 | 6.280 *** |
| | Small-scale | 909.75 | 687.60 | 222.15 | 32.31 | 15.477 *** |
| Shaanxi | Large-scale | 750.30 | 589.20 | 161.10 | 27.34 | 12.426 *** |
| | Small-scale | 984.60 | 679.65 | 304.95 | 44.87 | 25.241 *** |
| Gansu | Large-scale | 535.80 | 492.45 | 43.35 | 8.80 | 9.248 *** |
| | Small-scale | 607.35 | 533.40 | 73.95 | 13.86 | 10.571 *** |
| Ningxia | Large-scale | 495.15 | 454.35 | 40.80 | 8.98 | 5.982 *** |
| | Small-scale | 574.20 | 537.00 | 37.20 | 6.93 | 6.741 *** |
| Qinghai | Large-scale | 510.45 | 410.55 | 99.90 | 24.33 | 14.587 *** |
| | Small-scale | 619.65 | 485.40 | 134.25 | 27.66 | 16.246 *** |
| Xinjiang | Large-scale | 682.05 | 340.80 | 341.25 | 100.13 | 32.488 *** |
| | Small-scale | 759.30 | 358.35 | 400.95 | 111.89 | 35.260 *** |

Notes: *** $p < 0.01$ level of significance.

### 4.4. Influencing Factors of Fertilizer Use Deviation Degree

Next, we used OLS regression to empirically analyze the influencing factors of wheat farmers' fertilizer application amount and fertilizer application deviation degree. The results are shown in Table 7. First of all, the production scale of farmers had a significant negative impact on the amount of fertilizer application; in Column (1),the regression coefficient of risk was −6.315, which indicates that the large-scale farmers had less fertilizer application than small-scale farmers. In Column (3), the regression coefficient of farmers' scale to the deviation degree of fertilizer application was −5.4, which was highly significant at the significance level of 1%. This result shows that with the expansion of planting scale, the deviation degree of fertilizer application amount from the optimal application rate

gradually decreased. This is consistent with Hypothesis 1 that small-scale households have a higher deviation in fertilizer application and vice versa. The regression coefficients of risk aversion to fertilizer application amount and deviation degree in Column (1) and Column (3) were −2.31 and −2.145, respectively, which were significant at the significance level of 1%, indicating that risk averse farmers are more inclined to increase the amount of chemical fertilizer during wheat planting, and there was also a significant impact of risk aversion on deviation of fertilizer use, which verified Hypothesis 2, where the higher the risk aversion degree, the greater the deviation of chemical fertilizer application by the household.

**Table 7.** OLS regression results of influencing factors on fertilizer application.

| Variables | Actual Fertilizer Application (1) | Actual Fertilizer Application (2) | Deviation of Fertilizer Application (3) | Deviation of Fertilizer Application (4) |
|---|---|---|---|---|
| Scale | −6.315 *** | −5.310 *** | −5.403 *** | −9.375 *** |
| | (−5.78) | (−4.92) | (−4.82) | (−3.67) |
| Risk | −2.310 *** | −2.775 *** | −2.145 *** | −1.605 *** |
| | (−3.47) | (−3.29) | (−3.59) | (−3.25) |
| Rotation | 0.345 ** | 0.525 *** | 0.285 ** | 0.225 ** |
| | (1.98) | (2.41) | (2.09) | (2.03) |
| Scale*Risk | | −21.135** | | −18.075 *** |
| | | (−2.03) | | (−1.77) * |
| Site | 0.690 | −0.315 | 0.495 | 0.813 |
| | (1.26) | (−1.32) | (1.38) | (1.17) |
| Lot | 2.265 | 2.882 | 4.155 | 0.961 |
| | (1.48) | (1.65) * | (1.24) | (0.843) |
| Demonstration | −0.812** | −0.511 * | −0.393 | 0.855 |
| | (−1.99) | (−1.66) | (−1.41) | 1.22 |
| Gender | −0.885 | 0.255 | −0.691 | 0.336 |
| | (−1.33) | (0.97) | (−1.37) | (0.37) |
| Education | −3.915 *** | −4.830 *** | −2.895 *** | −1.141 * |
| | (−2.74) | (−2.96) | (−2.59) | (−1.59) |
| Years | −0.615 | 0.075 | −2.355 | −0.495 |
| | (−0.72) | (0.37) | (−0.69) | (−0.92) |
| Income | 5.431 | 9.825 | 3.825 | 6.752 |
| | (1.38) | (1.27) | (1.54) | (0.94) |
| Proportion | 3.165 *** | 2.295 ** | 2.310 *** | 1.144 ** |
| | (3.54) | (1.98) | (3.22) | (2.03) |
| Training | −0.375 ** | 0.211 * | −0.285 * | 0.105 |
| | (1.95) | (1.75) | (1.77) | (1.53) |
| Member | −0.735 * | −0.271 * | −0.242 | 0.495 |
| | (−1.69) | (−1.12) | (−1.03) | (0.86) |
| Area dummy variable | Provinces | Provinces | Provinces | Provinces |
| Constant | 21.692 *** | 26.955 *** | 19.935 *** | 31.441 ** |
| | (2.78) | (2.92) | (2.99) | (1.97) |
| No. of Obs | 780 | 780 | 780 | 780 |
| $R^2$ | 0.568 | 0.631 | 0.499 | 0.574 |
| F Test | 47.45 | 58.84 | 35.96 | 46.36 |

Notes: * $p < 0.10$, ** $p < 0.05$, *** $p < 0.01$ level of significance.

Whether crop rotation is carried out or not has a significant impact on the amount of chemical fertilizer applied to wheat planting. In the field of crop rotation, farmers tend to apply more chemical fertilizer. The reason is that due to the loss of chemical fertilizer and the decrease in soil fertility caused by multiple crop sowing, farmers tend to apply more chemical fertilizers to ensure the yield and quality of crops in the current season. Participation in agricultural high-tech demonstrations can effectively reduce the amount of chemical fertilizer applied by farmers. With the government and the public's attention to the environment, the national investment in research and development of environment-friendly agricultural technology has gradually increased, and has begun to

be promoted nationwide. Farmers participating in the demonstration of environmental friendly agricultural technology are more inclined to reduce the application of chemical fertilizers because they can make contact with advanced production technology and the environmental concept.

Additionally, we wanted to test the moderating effect of scale of cultivated land on the risk, examining whether the scale of cultivated land could mitigate the behavior of fertilizer application due to risk aversion by households. We introduced the intersection term Risk*Scale into the regression equation, and the regression result in Column (2) showed that the coefficient of the intersection term was −21.135, which was significant at the 1% level. This indicates that the increase in scale of land increases will mitigate the risk aversion by households in terms of fertilizer application. In Column (4), the coefficient of the intersection term was −18.075, which was significant at the 1% level. This indicates that the increase in scale of land increases will mitigate the risk aversion by households in terms of the deviation of fertilizer application. The results comply with Hypothesis 3.

Comparing Columns (1) and (3) with Columns (2) and (4), we see that the coefficient of the main regressors such as scale, risk, site, lot, etc. were still significant after applying the intersection term into the regression equation. Therefore, the regression model and conclusion are robust.

From the perspective of control variables, with the improvement in the wheat farmers' education level, then farmers' awareness of ecological protection was enhanced, and the amount and deviation degree of chemical fertilizer application decreased significantly. On the other hand, with the increase in the proportion of non-agricultural income in the total household income of wheat growers, farmers tended to use more chemical fertilizers instead of labor input to obtain a higher level of total household income. Therefore, the higher the proportion of non-agricultural income in the total household income, the greater the input of chemical fertilizer by wheat growers, and the higher the deviation of fertilizer application from the optimal level. The higher the enthusiasm of wheat growers to participate in wheat planting technology training, the more advanced technology they mastered, the stronger the awareness of environmental protection, and the lower the fertilizer investment and deviation degree.

## 5. Implication and Prospect

Agriculture is the foundation of the national economy, and fertilizer input is the basis of grain production. With the development of economy and the improvement in urbanization level, the grain industry is under the dual pressure of increasing production and efficiency as well as labor loss. Increasing the input of chemical fertilizers and other materials has become the main measure to alleviate this contradiction. However, excessive use of chemical fertilizer also brings a series of environmental problems such as soil fertility decline, groundwater pollution, and the like. Therefore, through the reform of the land system, promoting the large-scale operation of grain production has become a major policy to solve the problem of excessive use of chemical fertilizers in grain production in various regions of China.

In this paper, under the premise of fully considering the resource endowment and heterogeneity of small-scale farmers and large-scale households in production and management, different production decision-making frameworks for small-scale farmers and large-scale business households were constructed. Based on the wheat production provinces in northern China including Heilongjiang, Inner Mongolia, Shanxi, Shaanxi, Gansu, Ningxia, Qinghai, and Xinjiang, 300 large-scale growers and 480 small-scale farmers in eight provinces were selected. Based on the input–output data, this paper analyzed the fertilizer application and its influencing factors of small-scale and large-scale farmers in the northern wheat region, and further clarified whether the optimal fertilizer application amount and the development of scale management could solve the problem of excessive fertilizer use. The empirical results show that both small-scale farmers and large-scale farmers deviated from the optimal fertilizer application amount; moreover, the deviation

degree of optimal fertilizer application of small-scale farmers was significantly higher than that of large-scale farmers. In addition, not all the wheat growers in northern China have the problem of excessive use of chemical fertilizer. One of the reasons is that the large-scale promotion of the straw returning to field technology has replaced part of the chemical fertilizer input, which has reduced the usage amount of chemical fertilizers. Furthermore, the empirical results showed that when the risk aversion level was higher, farmers tended to apply more fertilizer to ensure grain output.

Reducing the amount of chemical fertilizer use and the deviation degree of chemical fertilizer application can not only reduce the farmers' production input, save production costs, and improve the farmers' income, but could also alleviate agricultural non-point source pollution caused by excessive application of chemical fertilizers and protect the ecological environment. Compared with small-scale farmers, the deviation degree of fertilizer application amount and optimal fertilizer application of large-scale farmers was relatively small, which indicates that the expansion of land management scale can significantly reduce the fertilizer input per unit area of wheat planting, which provides theoretical guidance and policy ideas for reducing the fertilizer application amount and controlling agricultural non-point source pollution in China. In future agricultural work, the government should continue to promote the reform of the land system, improve the land circulation system, develop moderate scale operations, and promote the transformation of China's agriculture from a small-scale agricultural economy to large-scale operation. In addition, the high risk aversion level of farmers will lead to the increase of fertilizer application amount and deviation degree. Therefore, it is of great significance to actively promote agricultural technology into households, improve the farmers' cognitive information of soil quality, and improve the agricultural insurance and subsidy system. In the survey, the local climate change and the original fertility heterogeneity were not taken into account, which may be time-variant factors that determine the amount and deviation of fertilizer used. In later work, more detailed content in the survey with more factors will be included to perfect the questionnaire and constructing model.

**Author Contributions:** Conceptualization, H.W. and Y.G.; methodology, H.W.; software, H.H.; validation, H.L. and H.S.; formal analysis, Y.G.; investigation, H.H. and Y.S.; resources, H.L.; data curation, H.S.; writing—original draft preparation, H.W.; writing—review and editing, Y.G.; visualization, Y.G.; supervision, H.W.; project administration, H.W.; funding acquisition, H.W. and Y.G. All authors have read and agreed to the published version of the manuscript.

**Funding:** This research was funded by "National Key Research and Development Program of China" "The Assessment of Application Effect of Fertilizer and Pesticide Reduction Technologies on Wheat in Northern China", grant number "2018YFD0200408"; "National Natural Science Foundation of China General Program "Economic Incentive, Peer Effects and Green Fertilization Behavior of Wheat Growers: A Randomized Controlled Trail", grant number "71973087"; National Natural Science Foundation of China Youth Fund Project "Effects of Incentive Heterogeneity on Agricultural Technology Extension: A Randomized Controlled Experiment in Northern Wheat of China", grant number "72003215"; The Ministry of education of Humanities and Social Science project "Ecological Compensation Mechanism of Protected Areas Based on Biodiversity Valuation", grant number "20YJC790114". Shaanxi Provincial Social Science Fundation "Research on Integration and Optimization of Shaanxi Agricultural Product Quality safety Traceability System from the Perspective of Circulation Innovation", grant number "2019D024"; The Special Fund project of Basic Scientific Research Operation funds of Central Universities: "The Influence of Peer Effect on Farmers' Green Fertilization Behavior: Based on the Analysis of the Randomized Controlled Experiment of Winter Wheat in Fen-Wei Plain", grant number "20SZYB21".

**Institutional Review Board Statement:** Not applicable.

**Informed Consent Statement:** Not applicable.

**Data Availability Statement:** Not applicable.

**Conflicts of Interest:** The authors declare no conflict of interest.

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
