# Peer review of "Farm Size, Risk Aversion and Overuse of Fertilizer: The Heterogeneity of Large-Scale and Small-Scale Wheat Farmers in Northern China"

_land, doi:10.3390/land10020111_

Round 1
Reviewer 1 Report
Dear authors,
I suggest the following comments/corrections that need to be improved if you want to be publish in Land.
Lots of English editing is necessary. If the English language is not significantly improved, the paper is not publishable.
You must also add some of the methodological limits of the empirical data and calculus method.
- CONTENT
Overall comment to authors : Results and analysis are interesting.
Page 1. Lines 40-41 : Excessive application of fertilizer directly led to the significant decline of fertility and fertilizer utilization rate of cultivated land in China.
Do you mean that excess application lead to a reduction of fertilizer use? Please be clearer.
Page 3. Line 117 : what do you mean by soil strength ??
Page 6, regarding survey data, what was the response rate ? How many farmers were questioned to have such a high number of responses ?
Page 8, results line 276-277 : Instead of "No matter the large-scale farmers or small-scale farmers" it would be clearer if you put "Whatever the farm size"
IMPORTANT METHODOLOGICAL ISSUE : It is essential that authors present the limits of this research. The statistics shown are regressions, but what are the limits of the validity of these data in terms of calculus, sample etc. ? This is important as no critical analysis is done in this article.
- FORM
English style must be improved, otherwise the article is unsuitable for publication and it will be rejected. Please ask this paper to be proof-read by a fluent English speaker.
I suggest some corrections in red to illustrate what needs to be done in all the text.
Some examples
Page 1. Line 26
Feeding a growing and increasing wealthy population is a grand major challenge.
Page 1, lines 32-34
Please put in two sentences. Always prefer short sentences in English.
Instead of : “In this context, in order to promote food supply 32 and ensure food security, the "high-input-high-output" agricultural production model has been 33 adopted by many regions and popularized on a large scale, the main feature of this model is to 34 increase the input of chemical fertilizer to achieve high output”
I suggest, I also suggest some corrections in red:
In this context, in order to promote food supply and ensure food security, the "high-input-high-output" agricultural production model has been adopted by many regions and popularized on a large scale. The main feature of this model is to increase the input of chemical fertilizers to increase production (high output)
Page 1, lines 34-35 some suggestions to put in the text. I put / to propose different versions of the same expression
the excess (or excessive) use of fertilizers/overuse of fertilizer/
Page 1, line 38 : kilograms per mu what does “mu” mean ? Please explain in a bracket.
Page 2, line 64 : Based on literature review and theoretical analysis,
Page 2, line 89 : “The logical structure of this paper is organized as follows” Why is this sentence here ? This must be corrected !
Page 2, line 93 : “the fifth part concludes”. I suggest : “, followed by a conclusion.”
Page 8, line 287 : Instead of "What’s more", I suggest a more formal expression such as "Moreover".
Please continue editing for the whole text.
Author Response
(1)Content and English writting: I have polished English writing.
(2)IMPORTANT METHODOLOGICAL ISSUE : It is essential that authors present the limits of this research. The statistics shown are regressions, but what are the limits of the validity of these data in terms of calculus, sample etc. ? This is important as no critical analysis is done in this article.
Answer: Thank you. In the last paragraph of the article, I have presented the limit of the research, In line 448-452 I added " In the survey, the local climate change and the original fertility heterogeneity are not taken into account, which may be time-variant factors that determine the amount and deviation of fertilizer used, in the later work, more detailed content in the survey more factors will be included for perfecting the questionnaire and constructing model."
(3) The meaning of "mu"
Page 1, line 38 : kilograms per mu what does “mu” mean ? Please explain in a bracket.
Answer: Thank you. "mu" is a Chinese unit used for measuring area, 1 mu=7175.94 square feet= 666.67 m^2. I have added the transformation of area unit into the paper so that it is understandable.
I have uploaded the newer version in the main page. Thank you.
Reviewer 2 Report
1. Line 363-371, this paragraph is not a summary, it should be put in the introduction. 2. kg/mu is not international units, should be "kg/ha". 3. Line 372-380, this section is repeated wirh the abstract and needs to be modified. 4. Quantitative data should be presented in the abstract, for example "the actual fertilizer application amount of small-scale farmers and large-scale farmers are 45.07 kg / mu and 36.70 kg / mu, respectively, while the optimal fertilizer application amount is 33.28 kg/ mu and 29.67 kg / mu,respectively, with deviation of 35.43% and 23.69%. (2) Not all the wheat growers in North China have the problem of excessive use of chemical fertilizer. Among them, the amount of chemical fertilizer applied by the wheat growers in Heilongjiang and Inner Mongolia did not reach the optimal level. The optimal level of chemical fertilizer application was 23.10 kg / mu and 22.35 kg / mu, while the 389 actual fertilizer use amount was 22.48 kg / mu and 21.64 kg / mu. "Author Response
Thank you very much.
Firstly I have updated the abstract, adding the key numbers. seconly in this paper, it contains a unit called "mu", which is a Chinese unit used for measuring area, 1 mu=7175.94 square feet= 666.67 m^2. I have added the transformation of area unit into the paper so that it is understandable.
I have uploaded a newer version on to the webpage. Thank you very much again.
Reviewer 3 Report
The paper assesses the relationship among the farm size, risk preference and fertilizer based on 300 large-scale and 480 small scale wheat farms in eight provinces in Northern China. The theme of the paper is very timely and interesting. Given the massive amount of fertilizer used in agricultural production in China, it is a balancing act to limit the fertilizer use and ensure the food security in China. Many studies argued the effective way to reduce the fertilizer use is to expand farm size because of the scale effect. The notable proponents of this idea include Ju et al. (2016) and Wu et al. (2018) as cited in this paper. A recent study (Xie et al., 2020) from a macro perspective published in Sustainability (https://www.mdpi.com/2071-1050/12/21/9299/htm) challenged that argument and argued there was no straight path yet for economy of scale effect yet for fertilizer use in China. Empirical studies like this paper would help understand the dilemma better and support the policy guideline moving forward. The additional dimension on the impacts of farmers’ risk preferences on fertilizer use would be also important for future agri-environmental policies in China
However, I found the methodology is questionable and the presentation is poor. On the methodology side, it was not clear how the optimal fertilizer application rate was defined. The formula to calculate the rate was given by Equation 2, but there was no support literature on how and why it is defined like that. Even for this definition, Pf and Py was not defined. If they were the prices of fertilizer and wheat, the formula might only define the economically optimal yield. I would argue that an optimal yield defined by the biophysical characteristics of the crop, local climate condition and yield would be more proper to assess the overuse or under use of fertilizers in agricultural production. Given the way the optimal yield is defined, the results and their implications would be invalid. There are so many issues on the presentation. The paper except the abstract is poorly written in English. It is so hard to follow and understand what the authors are trying to say. The authors should use the commonly used terms to describe certain phenomena in China instead of creating their own. There were many missing notations on equations and figures. Proper citations from literature should be used in the Methodology section to support the use of their methodology. The author should use the international unit standard instead of Chinese unit standards throughout the paper.
In summary, I liked the theme of the paper and also felt the data being collected to support this research was valuable. However, some key component of the methodology was wrong, and the presentation is too poor for publication.
Author Response
Firstly, the methodology is well developed as many scholar use the same or similar methods, in which I mentioned. Secondly, in this paper I didn't create new word or new term, I don't know why the reviewer 3 ask me not to create own word. As notations and equations , and citations of the figure in the have been included precisely. Third, the reviewer 3 doubts that there is no defination of "optimal fertilizer rate" this is calculated in Table 6 as it depends on various factor. So I doubt that if the reviewer 3 REALLY reads the paper? I admit that my english need improving, so I have polished the paper, making it readable. Compare with reviewer 1 ,2,and 4, who patiently point out the shortcoming and suggestion of the article, I really appreciate reviewer 1,2,4 's valuable advice, as for the reviewer 3's comment is really really makes me frustrate. I think that he simply write randomly as he has to subgmit the comment by deadline or if not, he is not professional in this field. So I sincerely hope that reviewer 3 can be substituted by another reviewer.
Reviewer 4 Report
This is an interesting study that deserve to be published. Overuse of fertilizers threatens the ecological environment. The authors discuss the impact of farm size and risk aversion on fertilizer’s overuse. The authors found that both small farmers and large farmers deviated from the optimal fertilization application amount, but the deviation of small farmers was higher than that of large farmers; the higher the risk aversion level, farmers tend to apply more fertilizer to ensure grain output. Thus, I recommend publishing it after minor revision. The specific comments are as follows:
(1) L1: I suggest that the author select the type of the paper.
(2) For the part of Results: The authors propose two hypotheses in L163-L169, but the authors seem to forget to tell the readers which results support or oppose these hypotheses.
(3) In table 4, there is a law of diminishing marginal returns in economics, and the input and output of fertilizers should be inverted “U”. I suggest that the author add a robustness test to show that the current production of Chinese wheat farmers has not exceeded the turning point.
(4) For the part of 4.4: I am more interested in the impact of the interaction of scale and risk on the amount of fertilization and the degree of fertilization deviation. The result of the interaction term helps to test Hypothesis 1.
(5) For the table 4 to 7: I suggest that the author explain the meanings of the numbers in brackets, and what do ***, **, * stand for?
Author Response
(1) L1: I suggest that the author select the type of the paper.
Thank you. I have updated this paper to "article"
(2) For the part of Results: The authors propose two hypotheses in L163-L169, but the authors seem to forget to tell the readers which results support or oppose these hypotheses.
Thank you . I have added the comment before Table 7, both Hypothesis 1 and 2 are support by the empirical part.
(3) In table 4, there is a law of diminishing marginal returns in economics, and the input and output of fertilizers should be inverted “U”. I suggest that the author add a robustness test to show that the current production of Chinese wheat farmers has not exceeded the turning point.
Thank you. In table 7, I added two column, which include the intersection of Risk*Scale I want to check if " The scale of cultivated land can mitigate the behavior of fertilizer application due to risk aversion by households." mentioned in Hypothesis 3. Column 2 and 4 does not only show that the scale has moderating effect on risk but the regression equation corresponding to Column 1 and 3 are robust as the significance of the confident of key variable are unchanged.
(4) For the part of 4.4: I am more interested in the impact of the interaction of scale and risk on the amount of fertilization and the degree of fertilization deviation. The result of the interaction term helps to test Hypothesis 1.
Thank you. In table 7, I included interaction term, putting it into Column 2 and 4.
(5) For the table 4 to 7: I suggest that the author explain the meanings of the numbers in brackets, and what do ***, **, * stand for?
Thank you I have updated them by adding notes.
The newer version can been seen from the uploaded file.